# Two Years of COVID-19 Pandemic: How the Brazilian Serie A Championship Was Affected by Home Advantage, Performance and Disciplinary Aspects

**DOI:** 10.3390/ijerph191610308

**Published:** 2022-08-19

**Authors:** Aldo Coelho Silva, Adriana Souza Amaral, Lucas Alves Facundo, Melissa Talita Wiprich, Leandro Rechenchosky, Wilson Rinaldi

**Affiliations:** 1Grupo de Estudos e Pesquisas Aplicadas em Futebol, Departamento de Educação Física, Universidade Estadual de Maringá, Maringá 87020-900, Brazil; 2Departamento de Educação Física, Universidade Federal de Minas Gerais, Belo Horizonte 31270-901, Brazil; 3Laboratório de Neuroquímica e Psicofarmacologia, Escola de Ciências da Saúde e da Vida, Pontífica Universidade Católica do Rio Grande do Sul, Porto Alegre 90619-900, Brazil

**Keywords:** home-field advantage, performance analysis, COVID-19, football, disciplinary aspects

## Abstract

As a result of the COVID-19 pandemic, the Brazilian Serie A championship was played without crowds in 2020 and partially in the 2021 season. We verified if the home advantage (HA) was different between the 2018, 2019, 2020, and 2021 seasons. We also compared the HA, performance, and disciplinary aspects between the rounds with or without crowds and verified the association between the number of absent athletes because of health protocols and the HA in the 2020 and 2021 seasons. We calculated the HA using the Pollard method. The performance aspects were goals, corners, shots, and ball possession, and the disciplinary aspects were fouls, yellow cards, and red cards. The HA was higher in the 2018 season compared with the other seasons. The rounds with crowds showed higher HAs than the two previous seasons and the teams had more shots and scored more goals than in the rounds without crowds. There were 457 athletes in the 2020 season and 123 athletes in the 2021 season who were absent because of health protocols, and there was no association between absence and HA. The COVID-19 pandemic affected soccer in the two last seasons in different ways in the Brazilian Serie A championship.

## 1. Introduction

A worldwide known sporting phenomenon is the home advantage. Specifically, in soccer, this phenomenon occurs when a team plays a friendly or competition in its stadium or field of command [1,2,3]. Pollard [4] provided the most commonly used method to calculate home advantage, which expresses the percentage of points obtained or scored at home in terms of the respective total number of points obtained or total goals scored [4]. A home advantage is represented numerically when more than 50% of the total points are obtained in matches in which the team plays at home, whereas values below 50% do not indicate an advantage. Some factors can explain home advantages, such as the deleterious effect of travel fatigue for the away team, the familiarity with local conditions, the support of the home team’s crowd, and the quality of the team [5,6,7]. Ponzo and Scoppa [8] analyzed the home advantage only in derby matches, isolating the familiarity and fatigue factors. The authors observed a greater home advantage and greater amount of goals scored from the home teams, possibly due to the support and motivation of the crowd for the athletes. Changes in the physiological component can help to explain the better performance of teams in their domains. For example, the testosterone levels of soccer players may be higher in the pre-game of home matches than in the pre-game away from home. Added to the motivation from crowd support, this higher level of testosterone, the greater aggressivity, and the perception of territorial dominance can serve as the basis for the better performance by players when matches are played at home [9,10]. Regarding crowd support, athletes feel more motivated when they play at home and with a full stadium, resulting in more energy and effort for actions taken in the match [11].

At the end of 2019, the world was informed of the emergence of a virus, from the coronavirus family, which mainly affected the respiratory system. This new coronavirus was named as COVID-19 or SARS-CoV-2 [12]. In March 2020, the World Health Organization declared a global pandemic [13]. Most elements of competitions in progress during that period were interrupted or ended [14]. After the elaboration of health protocols, several competitions returned from the round in which they were interrupted. A common feature of the health protocols of different leagues around the world was the return to competitions without the presence of crowds. In this way, the possible effect of the presence of crowds was nullified, which could affect the home advantage. The first studies on home advantage during the COVID-19 pandemic showed contradictory results [15]. Some studies showed a lower home advantage in the points obtained in the rounds following the return to competitions, while other studies did not find any change in HA [15,16,17,18]. Besides the analysis of home advantage in the points obtained, other studies compare home advantage in physical performance [19,20] and the disciplinary bias between the rounds without and with crowds. Likewise, contradictory results were presented.

Brazil is the nation that has won the most FIFA World Cups and has the highest number of players who have won the FIFA World Player of The Year. The Brazilian Serie A championship traditionally starts in May, with the final rounds taking place in December. In 2020, the Brazilian Football Confederation (CBF) determined that the athletes’ training was interrupted, which occurred in March. As the months progressed and after meetings between several members of the confederation and clubs, the CBF prepared a health protocol that allowed the gradual return of athletes to training and competitions. Because of the COVID-19 pandemic in 2020, the Brazilian Serie A Championship started in August and ended in 2021 and, for the first time in history, was held completely without crowds. In the following season (starting in 2021), the Brazilian Serie A Championship started in May and ended in December. However, health protocols requiring matches without crowds in stadiums because of the COVID-19 pandemic were changed in round 22, with the remaining 16 rounds held with crowds. In this way, the Brazilian Serie A Championship allows for a unique analysis of HA, with the comparison between the full season without crowds and a season partially without crowds as a result of the COVID-19 pandemic.

A reduction in home advantage has been observed in the main world leagues in recent years, and in some cases, even with the closing of the stadiums to crowds, no significant change was observed in the advantage [16]. With the COVID-19 pandemic, there is the possibility of verifying if there is a tendency to reduce the home advantage in the Brazilian Serie A Championship and the real role of the crowd in the home advantage. Analyzing this trend in Brazilian soccer is important to guide teams, coaches, and staff about the challenges that home matches can provide. According to the presented research, our initial hypothesis was that the 2020 and 2021 seasons would have the lowest home advantage in terms of points obtained compared with the 2018 and 2019 seasons; that the home advantage in terms of points obtained, the home advantage in terms of performance, and disciplinary aspects of the seasons without crowds would be worse than those with crowds; and that there would be associations between the number of absent athletes because of health protocols and the home advantage in terms of points obtained in the 2020 and 2021 seasons. Thus, we aimed to verify if there was a difference in the home advantage in terms of points obtained between the last four seasons (2018, 2019, 2020, and 2021) of the Brazilian Serie A Championship. We also aimed to verify if there was a difference in the home advantage in terms of points obtained, the home advantage in terms of performance, and disciplinary aspects between the rounds with and without crowds in the last two seasons (2020 and 2021). Then, we aimed to describe the number of absent athletes because of the mandatory health protocol and tested the association between home advantage in terms of points obtained and the number of absent athletes because of the health protocol.

## 2. Materials and Methods

The championship data from the last four seasons (2018, 2019, 2020, and 2021) were obtained from https://www.cbf.com.br/futebol-brasileiro/competicoes/campeonato-brasileiro-serie-a, the official website of the CBF (accessed on 20 December 2021). The number of athletes absent because of the health protocol was obtained from each club’s website for each round.

### 2.1. Quantifying the Home Advantage

The home advantage in terms of points obtained is equal to the points obtained in matches played at home, divided by the sum of the points obtained at home and away from home, and multiplied by 100 (HA = H/H + A × 100). In this method [3,4], the value is always positive, and a value of 50% indicates the balance in the points obtained at home and away from home, while values above 50% indicate an advantage in playing at home, and values below 50% indicate that there is no advantage in playing at home. The same method quantified the HA for the performance and disciplinary aspects.

### 2.2. Performance and Disciplinary Aspects

The performance aspects considered were goals, corners, shots, and ball possession. The disciplinary aspects considered were fouls, yellow cards, and red cards. Data for each aspect were obtained match by match in the 2020 and 2021 seasons and collected on the website www.transfermarkt.com.br (accessed on 26 December 2021). This website has been used by researchers to obtain the results of several championships during the 2020 and 2021 seasons to determine the home advantage [15,16].

### 2.3. Statistical Analyses

The Kolmogorov–Smirnov test analyzed the distribution of data. The mean and standard deviation represented the values of central tendency and dispersion, respectively, of the variables with a parametric distribution. The median and interquartile range represented the measures of central tendency and dispersion, respectively, of the variables with a non-parametric distribution. The Levene test evaluated the equality of variances of home advantage in terms of points obtained in all seasons. Then, one-way analysis of variance (ANOVA) verified the differences in home advantage in terms of points obtained between the seasons. When ANOVA showed a difference between the seasons, Tukey’s post hoc was used to identify the differences. We performed two analyses on the conditions with and without crowds concerning the performance and disciplinary aspects. We have compiled the entire 2020 season and the first part of the 2021 season containing the first 22 rounds (2021a) to describe the condition without crowds and the second part of the 2021 season containing matches from round 23 through round 38 (2021b). In this analysis, Welch’s test for unequal variances compared the conditions regarding the performance and disciplinary aspects. The eta partial squared described the effect size in the comparisons between the seasons and the Cohen’s d adjusted to unequal sample size described the effect size with a 95% confident interval in the two comparisons between the conditions with and without crowds. The effect size classification by partial eta square was small = 0.01 to 0.05, medium = 0.06 to 0.13, and large ≥ 0.14, while for the Cohen’s d, it was trivial ≤ 0.2, small > 0.2 to 0.4, medium > 0.5 to 0.7, and large ≥ 0.8 [21]. Spearman’s correlation test analyzed the association between home advantage in the points obtained and the number of absent athletes because of the health protocol. The association’s classification was trivial ≤ 0.2, small > 0.2 to 0.4, medium > 0.5 to 0.7, and large ≥ 0.8 [22]. A *p*-value < 0.05 was considered to reject the null hypothesis in all analyses.

## 3. Results

Table 1 shows the home advantage in terms of points obtained for the 2018, 2019, 2020, and 2021 seasons. There was a statistical difference between the seasons. The home advantage in terms of points obtained was higher in the 2018 season compared with the 2019 (*p* = 0.002), 2020 (*p* = 0.001), and 2021 seasons (*p* = 0.001). There was no difference between the other seasons in any analyses.

In the second analysis of home advantage in terms of points obtained over the seasons, there was a difference between the seasons (Table 2). The home advantage in terms of points obtained was higher in the 2018 season compared with the 2020 (*p* = 0.018) and 2021a seasons (*p* = 0.001). The home advantage in terms of points obtained for the 2021b season was higher than that of the 2019 (*p* = 0.001), 2020 (*p* = 0.001), and 2021a (*p* = 0.001) seasons. The 2020 and 2021a seasons together had 602 matches, where the home teams won 254 matches (42.2%) and the away won 165 matches (27.4%). In 2021b, the number of matches won by the home teams increased. There were 158 matches, with 91 victories by home teams (57.6%) and 29 victories by away teams (18.4%). Regarding the first analysis between the conditions with and without crowds, the descriptive and comparative data of home advantage in terms of points obtained, performance, and disciplinary aspects are shown in Table 2. During the rounds with crowds, the home advantage in terms of points obtained as well as the home advantage for shots and goals were higher, and the home advantage for faults was lower than in the rounds without crowds. Moreover, during the rounds with crowds, the home teams achieved more points, had more shots, and received more corners. On the other side, the away teams achieved fewer points and scored fewer goals.

Concerning the second analysis between the conditions with and without crowds, the descriptive and comparative data of home advantage in terms of points obtained, performance aspects, and disciplinary aspects are shown in Table 3. The home advantage in terms of points obtained and the home advantage for goals were higher in rounds with crowds than in the rounds without crowds. In the last 16 rounds, the home teams achieved more points, had more shots, and received more corners, while the away teams achieved fewer points and scored fewer goals.

Figure 1 shows the number of athletes absent because of the health protocol during the 2020 and 2021 seasons. Absence due to health protocol occurred for 457 athletes in the 2020 season (November had the highest number of monthly absences, with 170 absences), while in the 2021 season, 123 athletes were absent (June had the highest number of monthly absences, with 75 absences). The Spearman test did not identify statistical significance in the association between home advantage in the points obtained in the 2020 season and the number of players on the home team (R = −0.053; *p* = 0.152) or away team (R = −0.008; *p* = 0.442) who were absent because of the health protocol. The analysis of each round identified an association between home advantage in the points obtained and the number of players on the away team who were absent because of the health protocol in rounds 24 (R = 0.562; *p* = 0.045) and 29 (R = −0.566; *p* = 0.044). Round 24 took place in December (2 to 7 December) and round 29 took place in January (6 and 7 January). Spearman’s test did not identify statistical significance in the association between home advantage in the points obtained in the 2021 season and the number of players on the home team (R = −0.032; *p* = 0.272) or away team (R = −0.077; *p* = 0.070) who were absent because of the health protocol. The analysis of each round identified an association between home advantage in terms of points obtained and the number of players from the away team who were absent because of the health protocol in round 4 (R = −0.581; *p* = 0.039), which took place in June (16 and 17 June) and November (2 November).

## 4. Discussion

We initially aimed to verify if there was a difference in home advantage in terms of points obtained between the last four seasons (2018, 2019, 2020, and 2021) of the Brazilian Serie A Championship, and in a second analysis, with the 2021 season divided into the first part without crowds and the second part with crowds. Our first hypothesis was partly confirmed as we see that the home advantage in terms of points obtained for the 2020 and 2021 seasons was lower than the 2018 season, but not than the 2019 season. Our second hypothesis was partly confirmed, as we see that the home advantage in terms of points obtained, as well as some performance and disciplinary aspects, were worse in rounds without crowds when compared with the last 16 rounds of the 2021 season performed with crowds. Lastly, our third hypothesis was refuted as we did not see an association between the number of athletes absent because of the health protocol and the home advantage in terms of points obtained.

Initially, the expectation would be a reduction in the home advantage over the seasons. In our first analysis, we observed that, in all seasons analyzed, regardless of the crowd support, the mean of home advantage was always higher. Besides, there was a difference statistically significant and large effect size in the decreases in home advantage in terms of points obtained, with the 2018 season showing the highest home advantage in terms of points obtained. However, the statement that there is a tendency to reduce home advantage in terms of points obtained in the Brazilian Serie A championship should be viewed with caution. In the second analysis of the patter of home advantage in the points obtained in the last four seasons, when we divide the 2021 season into a part without (2021a) and a part with crowds (2021b), we observed a large effect size on home advantage in the points obtained with a significant increase. When using the 38 rounds in the 2021 season, the home advantage in the points obtained was 61.4%, while the home advantage in the points obtained when the rounds were split was 54% for rounds without crowds and 75.4% for rounds with crowds, leading to a U-shaped home advantage in the pattern of points obtained. The increase in home advantage in the points obtained in the 2021b can be explored further with the analysis of rounds with or without crowds in the last seasons. The home advantage in terms of points obtained, the home advantage for shots and goals (first analysis), and the home advantage in terms of points obtained and home advantage for goals (second analysis) were higher with the crowd presence. Moreover, in rounds with crowds, the home teams won almost 28% of the games, while in the rounds without crowds, this number dropped to almost 10%. On the other hand, the other performance variables were not different with the return of the crowd to the stadiums.

In the last two years, several studies have investigated the effect of matches without crowds because of the COVID-19 pandemic on performance variables in European leagues, with some contradictory results. Wunderlich, Weigelt [23] did not observe a reduction in the home advantage in the rounds without crowd presence in 11 European leagues. Similarly, the last 10 matches without crowds in the Portuguese league did not affect home advantage in terms of points obtained [18]. On the other hand, McCarrick, Bilalic [24] analyzed 15 leagues in 11 countries. The authors noted that the home advantages in terms of points obtained and goals scored were reduced during the COVID-19 pandemic. The authors reported that, in matches without crowds, the performance of the home team was lower, with few offensive actions, in addition to the disciplinary bias being reduced (number of fouls and yellow cards). Scoppa [25] observed that home teams showed reductions in the achievement of points gained, goals, finishes, shots on target, and corners in the ten European leagues analyzed. Correia-Oliveira and Andrade-Souza [16] observed a reduction in home advantage in the two main divisions of the English league, the first division of the German league, and the Spanish LaLiga^TM^. Interestingly, without crowd support, the home advantage of the German and English leagues was below 50%. In the 2019/2020 Bundesliga, there was a 28% reduction in the number of home teams’ victories and a 44% increase in the number of away teams’ victories when the crowd was prevented from attending the stadiums from the 26th round. The home advantage in the points obtained was lower (44.1%) without crowds when compared with the 25 rounds with crowds (54.3%) [17,20].

The comparison between the rounds with or without crowds on the disciplinary aspect showed a reduction in home advantage for faults, though this effect size was trivial. The analyses in the 2021 season did not show significant differences between the two parties within the disciplinary aspect. There was no home advantage for yellow cards, red cards, or fouls. We can speculate that the presence of technology such as the VAR may have aided in the refereeing team’s decisions and not randomly punishing away teams [26]. Our results differ from what recent studies on the effect of the pandemic on the disciplinary aspect have presented. In the Bundesliga, when the championship returned without crowds, the home advantage for yellow cards and the number of fouls was reduced, while in matches with crowds, the distribution of yellow and red cards was greater for home and away teams. In the 15 leagues analyzed, McCarrick, Bilalic [24] found that the disciplinary bias was reduced without crowds. Reade, Schreyer [27] analyzed the results of seven European competitions since the 2002/2003 season and observed that the discipline bias was reduced, highlighting that away teams had a reduction in the number of yellow cards received and the number of fouls committed. Additionally, a recent study showed a reduction in referee bias without crowds in 12 different leagues and reported that this bias constitutes 20% of the home advantage index reduction during the pandemic [28]. Collectively, these results suggest that crowds influence refereeing team decisions, increasing the number of fouls and cards applied to away teams. The social pressure evoked by the crowd in the stadiums causes bias in the interpretation of actions carried out by the home and away teams.

The return of crowds to stadiums was different in all of the studies presented; consequently, the number of rounds without crowds was different. This fact must be considered. As there is a difference between performance and disciplinary aspects when crowds were present or not in the stadiums, it is possible that, the longer the duration of the rounds without crowds, the greater the effect of the COVID-19 pandemic on the final classification in of the various leagues analyzed. For example, the matches without crowds in the Bundesliga started on the 26th round out of 34 rounds, LaLiga^TM^ began on the 28th round out of 38 rounds, and the Premier League started on the 30th round out of 38 rounds. In Brazil, it was the opposite. The entire 2020 season took place without crowds and, in the 2021 season, the crowd returned only in the last 16 rounds, with almost 60% of the championship played without crowds in stadiums. In this way, the teams in the Brazilian Serie A Championship may have been more affected, considering the long period without crowds in the stadiums. It is essential to highlight that the difference between the champion and second place of the 2020 season was 1 point and the difference between those who qualified or not for the CONMEBOL Libertadores Cup was 2 points in 2020 and 1 point in 2021; concerning relegation, in the 2020 season, relegation was decided on goal difference and, in the 2021 season, relegation was decided by a difference of 3 points. When analyzing the differences in performance aspects between the rounds with and without crowds and with the final classification of the two seasons impacted by the COVID-19 pandemic, we can observe that the pandemic directly affected the classification of teams in the championship.

Another important point to be discussed in our results is the peaks of COVID-19 cases in November 2020 (170 cases), January 2021 (80 cases), and June 2021 (75 cases) in the Brazilian Serie A Championship. The peaks of cases manifested in the competition reflected the peak of the disease’s first and second waves in Brazil, according to the Brazilian Health Ministry [29]. In June 2020, 887,841 new positive tests were recorded, with a daily average of 29,594.7 infections. In 2021, there were 1,528,758 cases and an average of 49,314.8 infections in January. In June, 2,011,587 infections were recorded, with an average of 67,052.9 infections per day. Even with the start of vaccination in Brazil in January 2021, only 22.64% of the population was vaccinated in June, which is still a small rate to control the endemic state of COVID-19 [30]. It is important to note that COVID-19 does not only keep athletes away from teams, but also professionals from the coaching staff, who may have more severe cases of the disease and even death [31].

Additionally, a reduction in infections was noted between the 2020 and 2021 seasons. In 2021, vaccination against COVID-19 began in Brazil, but at a slow pace, as previously reported; only after June did this process accelerate, which was the first time that more than 1 million people were vaccinated in one day [30]. Throughout 2021, more than 143 million people were vaccinated with both doses (complete vaccination schedule) in Brazil, totaling about 67% of the population in the country. The advancement of immunization played a key role in this process, as vaccines proved to be effective and safe against the disease [32]. As vaccination advanced, cases and deaths from COVID-19 in the country decreased [29], which also reduced cases in competition. It is important to highlight that the curve of infections was similar to the cases in other countries. In observations of an Italian top-level team, in the early waves of the pandemic (May 2020), a higher rate of COVID-19 was observed, as well as a reduction in cases in February of 2021. Similarly, our data showed a higher number of cases in August of 2020 than in February of 2021 [33]. Is important to note that the number of cases was impacted by the protocols adopted by the clubs and federations, and the development of strategies such as continuous testing, physical distancing, and strict hygiene protocols (masks) could mitigate the infections in the soccer clubs and leagues [34].

Another interesting result is the absence of a change in home advantage in the points obtained in matches with athletes absent because of the health protocol. Nowadays, teams have become increasingly balanced in physical and technical-tactical aspects, in addition to having more data on opponents [11]. With more information-based planning, it is possible to devise strategies that can circumvent the opposing team’s strengths and obtain information on the location where the match will be held (e.g., state of the pitch and altitude) [11]. The same occurs in cases of the removal of athletes, as with more information about the athletes of your team and your opponent, it will be possible to circumvent the removal more assertively. Another important factor that could have impacted home advantage is the quality of the teams, as more qualified teams are around 5% more likely to win matches at home [6], which could also affect matches played away from home, reducing the chances of defeat. At this point, more qualified teams would also have a more complete roster and reserve athletes that can effectively cover the absences. Different impacts of the number of players out as a result of the COVID-19 infection in the analyses of home advantage for each round were observed in the 2020 and 2021 seasons. In two rounds of the 2020 season (24 and 29), opposite results were presented, with negative and positive influences between home advantage and the number of players on the away team absent because of the disease. This discrepancy between this round could be because the number of players infected in January (80 cases) was double that in December (40 cases). In round 29 (January), the higher number of infections could remove important players from both the home and away teams. Despite the evolution in the modality and tactical aspects [11], some results in soccer could be impacted by some players’ quality and characteristics owing to the specific strategy and tactics of the coaches. In this way, if key players are out because of viral infection, the results and, consequently, the home advantage could be impacted, which could explain the differences between each round. In the 2021 season, a negative correlation was observed in round 4 between players out because of COVID-19 infection and home advantage. In this case, most of the matches were played in June, which was the month with more cases in the Brazilian Serie A Championship and fewer athletes available. Most of the time, away teams need to travel to play matches, and this constant travel could impact away team performance [35]. This factor with fewer players available could reinforce the home advantage in this round.

This article has some limitations that should be considered when analyzing the results. The home advantage was not analyzed with a variable that could adjust the analysis according to the athlete’s quality. In addition, it was not possible to describe whether the athletes absent because of COVID-19 were starting players. Despite that, this article shows important information that should be considered by the clubs and the sports scientists. Between the years 2020 and 2022, the world was informed of COVID-19 mutations, leading to an increase in infected people, deaths, and new restrictive measures [36]. Currently, China has reported a lockdown in the city of Shanghai, concomitantly with a new mutation of the virus, called BA.2 [37]. This article contributes to expanding the field of sports science, providing information in the analysis of the performance of professional soccer teams that goes beyond psychological, technical, and tactical factors, by indicating that extra-field factors, such as the pandemic, directly impact sports performance. The present study can help coaches and staff by indicating that they have to pay attention to the effect of external factors such the presence of crowds. Once the presence of crowds can be limited, there can be a reduction in the home advantage, and the preparation of visiting teams can be reviewed. For example, in situations where there is no crowd, the home team has fewer offensive actions, such as corners or shots, which may allow the away team’s command to organize its team with a more offensive tactic.

## 5. Conclusions

The current study provides information about the impact that the COVID-19 pandemic has had on the Brazilian Serie A Championship in the last two seasons. Moreover, despite of the importance of Brazil to world soccer, this is the first article dealing with the effects of the COVID-19 pandemic on home advantage in the entire two seasons played out with a rigid health protocol in the Brazilian Serie A championship. The reduction in the home advantage in the 2020 season compared with the 2018 season has affected the results of the teams in the dispute of several objectives during the competition, a fact that occurred again in the 2021 season. When the crowd was allowed to return to the stadiums, even with a reduced amount in terms of percentage of occupation, the home advantage increased; however, this increase may have been at a time in the competition at which several teams had already lost points as the home team that, in a racing points championship, were decisive in the final rounds of the competition.

## Figures and Tables

**Figure 1 ijerph-19-10308-f001:**
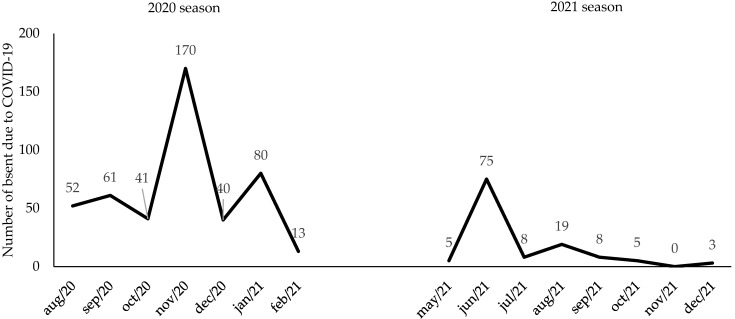
Soccer players absent because of the health protocol in the 2020 and 2021 seasons in the Brazilian Serie A championship.

**Table 1 ijerph-19-10308-t001:** Comparison of the home advantage between the last four seasons of the Brazilian Serie A championship.

	Mean ± SD	*f*-Value	*p*-Value	Effect Size (Classification)
First analysis				
2018	70.7 ± 6.7 *	10.05	0.00	0.28 (large)
2019	62.7 ± 6.2			
2020	60.4 ± 5.5			
2021	61.4 ± 7.8			
Second analysis				
2018	70.7 ± 6.7 *	13.82	0.00	0.36 (large)
2019	62.7 ± 6.2			
2020	60.4 ± 5.5			
2021a	54.0 ± 13.6			
2021b	75.4 ± 14.7 ^#^			

First analysis: * statistical difference from the 2019, 2020, and 2021 seasons. Second analysis: ^#^ statistical difference from the 2019, 2020, and 2021a seasons.

**Table 2 ijerph-19-10308-t002:** Statistical and descriptive data across two seasons over the COVID-19 pandemic in the Brazilian Serie A championship.

		Without Crowd			With Crowd			*t*-Value	*p*-Value	Effect Size	Confident Interval	Effect Size Classification
Points	H	1.57	±	1.20	1.97	±	1.20	−3.55	0.00 *	−0.33	−0.50/−0.15	Small
	A	1.13	±	1.22	0.79	±	1.12	3.26	0.00 *	0.28	0.10/0.45	Small
Goals	H	1.34	±	1.09	1.39	±	1.08	−0.52	0.60	−0.04	−0.22/0.12	Trivial
	A	1.06	±	1.02	0.82	±	1.00	2.67	0.00 *	0.23	0.06/0.41	Small
Shot	H	13.55	±	5.01	15.06	±	5.12	−3.31	0.00 *	−0.29	−0.47/−0.12	Small
	A	11.30	±	4.37	11.20	±	4.85	0.22	0.82	0.02	−0.15/0.19	Trivial
Ball possession	H	51.46	±	10.43	51.01	±	10.22	0.49	0.62	0.04	−0.13/0.21	Trivial
	A	48.53	±	10.43	48.92	±	10.20	−0.43	0.66	−0.03	−0.21/0.13	Trivial
Corners	H	5.42	±	2.89	6.06	±	3.20	−2.26	0.02 *	−0.21	−0.39/−0.04	Small
	A	4.68	±	2.69	4.25	±	2.39	1.96	0.05	0.16	−0.01/0.33	Trivial
Yellow card	H	2.26	±	1.39	2.23	±	1.42	0.22	0.82	0.02	−0.15/0.19	Trivial
	A	2.18	±	1.45	2.40	±	1.50	−1.65	0.09	−0.15	−0.32/0.02	Trivial
Red cards	H	0.13	±	0.36	0.09	±	0.28	1.39	0.16	0.11	−0.05/0.29	Trivial
	A	0.12	±	0.36	0.11	±	0.33	0.35	0.72	0.02	−0.14/0.20	Trivial
Faults	H	15.55	±	4.30	15.08	±	4.36	1.19	0.23	0.10	−0.06/0.28	Trivial
	A	15.05	±	4.29	14.67	±	4.58	0.94	0.34	0.08	−0.08/0.26	Trivial
HA_points_ (%)		57.39	±	41.08	69.62	±	39.03	−3.46	0.00 *	−0.30	−0.47/-0.12	Small
HA Goals (%)		55.65	±	34.08	67.44	±	34.86	−3.60	0.00 *	−0.34	−0.52/-0.16	Small
HA Shot (%)		27.39	±	42.77	57.51	±	14.82	−14.21	0.00 *	−0.77	−0.95/-0.59	Moderate
HA Ball possession (%)		51.46	±	10.43	51.04	±	10.20	0.46	0.64	0.04	−0.13/0.21	Trivial
HA Corners (%)		51.22	±	91.35	57.68	±	21.58	−1.56	0.11	−0.07	−0.25/0.09	Trivial
HA Yellow card (%)		51.43	±	25.21	47.81	±	22.37	1.75	0.08	0.14	−0.02/0.32	Trivial
HA Red cards (%)		50.27	±	47.04	43.33	±	44.09	0.70	0.48	0.14	−0.02/0.32	Trivial
HA Faults (%)		67.79	±	156.44	50.75	±	9.95	2.61	0.00 *	0.12	−0.05/0.29	Trivial

* Statistical difference between the conditions. HA = home advantage. HA_points_ = home advantage for points obtained.

**Table 3 ijerph-19-10308-t003:** Statistical and descriptive data for the 2021 season are divided into parts with and without crowds in the Brazilian Serie A championship.

		Without Crowd			With Crowd			*t*-Value	*p*-Value	Effect Size	Confident Interval	Effect Size Classification
Points	H	1.46	±	1.20	1.97	±	1.20	−3.83	0.00 *	−0.42	−0.63/−0.21	Small
	A	1.19	±	1.22	0.79	±	1.12	3.44	0.00 *	0.33	0.13/0.54	Small
Goals	H	1.19	±	1.06	1.39	±	1.08	−1.71	0.08	−0.18	−0.39/0.01	Trivial
	A	1.03	±	0.97	0.82	±	1.00	2.07	0.03 *	0.21	0.00/0.41	Small
Shot	H	13.61	±	4.61	15.06	±	5.12	−2.82	0.00 *	−0.29	−0.50/−0.09	Small
	A	11.08	±	4.23	11.20	±	4.85	−0.25	0.79	0.02	−0.23/0.17	Trivial
Ball possession	H	51.96	±	9.98	51.01	±	10.22	0.89	0.37	0.09	−0.10/0.29	Trivial
	A	48.04	±	9.98	48.92	±	10.20	−0.83	0.40	−0.08	−0.29/0.11	Trivial
Corners	H	5.44	±	2.65	6.06	±	3.20	−1.98	0.04 *	−0.21	−0.41/0.00	Small
	A	4.66	±	2.68	4.25	±	2.39	1.55	1.21	0.15	−0.04/0.36	Trivial
Yellow card	H	2.25	±	1.32	2.23	±	1.42	−1.16	0.86	0.01	−0.18/0.21	Trivial
	A	2.19	±	1.44	2.40	±	1.50	−1.36	0.17	−0.14	−0.34/0.06	Trivial
Red cards	H	0.13	±	0.35	0.09	±	0.28	1.13	0.25	0.12	0.08/0.32	Trivial
	A	0.09	±	0.32	0.11	±	0.33	−0.81	0.41	−0.06	−0.26/0.14	Trivial
Faults	H	15.30	±	3.99	15.08	±	4.36	0.50	0.61	0.05	−0.15/0.25	Trivial
	A	14.71	±	4.06	14.67	±	4.58	0.85	0.93	0.00	−0.19/0.21	Trivial
HA_points_ (%)		54.50	±	40.66	69.62	±	39.03	−3.65	0.00 *	−0.37	−0.58/−0.16	Small
HA Goals (%)		52.96	±	34.73	67.44	±	34.86	−3.77	0.00 *	−0.41	−0.63/−0.19	Small
HA Shot (%)		55.25	±	13.61	57.51	±	14.82	−1.50	0.13	−0.15	−0.36/0.04	Trivial
HA Ball possession (%)		51.95	±	9.98	51.04	±	10.20	0.86	0.38	0.09	−0.11/0.29	Trivial
HA Corners (%)		54.17	±	20.79	57.68	±	21.58	−1.58	0.11	−0.16	−0.37/0.03	Trivial
HA Yellow card (%)		51.38	±	24.11	47.81	±	22.37	1.47	0.14	0.16	−0.03/0.37	Trivial
HA Red cards (%)		62.39	±	47.09	43.33	±	44.09	1.64	0.10	0.40	−0.09/0.92	Small
HA Faults (%)		51.01	±	9.21	50.75	±	9.95	0.26	0.79	0.02	−0.18/0.24	Trivial

* Statistical difference between the conditions. HA = home advantage. HA_points_ = home advantage for points obtained.

## Data Availability

No new data were created in this study. Data sharing is not applicable to this article. The data are publicly accessible on the follow websites: Brazilian Football Confederation (www.cbf.com.br, accessed on 20 December 2021) and Transfer market (www.transfermarkt.com.br, accessed on 26 December 2021). The number of athletes absent because of the health protocol was obtained from each club’s website for each round. If you need more details, please contact the corresponding author.

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
