# Peer review of "Two Years of COVID-19 Pandemic: How the Brazilian Serie A Championship Was Affected by Home Advantage, Performance and Disciplinary Aspects"

_ijerph, 2022, doi:10.3390/ijerph191610308_

Round 1
Reviewer 1 Report
This paper is a creative paper with an interesting topic.
Please check some items to make this paper a better paper.
1. In this paper, it is said that if it exceeds 50%, there is generally a home advantage. Therefore, the data presented in this paper shows that in most cases, the figure is over 50%
Although there is a statistical difference from the previous season, I hope the author's interpretation of whether home adventure can be said to be due to without crowd.
2. The results of this paper specify the results of the hypothesis. No content was found for the hypothesis in the body. Please present the hypothesis in the text.
3. There is a typo in Table 3. Whit->Please correct it.
4. Contribution to the field is omitted
Author Response
Dear Reviewer, I thank you for your patience and appreciable comments that have been so helpful in improving the manuscript’s quality. Just below are responses to these comments. I have updated the version of my manuscript, so you can check the new version in which I have taken into account your comments. I hope you will agree that this new version is better than the previous one.
- In this paper, it is said that if it exceeds 50%, there is generally a home advantage. Therefore, the data presented in this paper shows that in most cases, the figure is over 50%
Although there is a statistical difference from the previous season, I hope the author's interpretation of whether home adventure can be said to be due to without crowd.
Answer: We agree with the reviewer and now we indicated in the discussion that, regardless of the presence of the crowd, in the Brazilian soccer championship there is a home advantage. We present this information on page 12, line 197: “We observed that in all seasons analyzed, regardless of the presence or absence of the crowd, the mean of HÁ was always above 50%.”
- The results of this paper specify the results of the hypothesis. No content was found for the hypothesis in the body. Please present the hypothesis in the text.
Answer: We apologize for not being clear in presenting the research hypotheses. We had written the hypothesis in the middle of the third paragraph on page 2. However, we have now moved the hypothesis so that it is at the beginning of the paragraph and makes identification easier for the reader. Page 4 line 75 “According to the presented, our initial hypothesis was that the 2020 and 2021 seasons would have the lowest home advantage in the points obtained compared to the 2018 and 2019 seasons, the home advantage in the points obtained, the home advantage of the performance and disciplinary aspects of the seasons without the crowd would be worse than those with the crowd and that there would be associations between the number of athletes absent due to health protocol and the home advantage in the points obtained in the 2020 and 2021 seasons”
- There is a typo in Table 3. Whit->Please correct it.
Answer: Initially, we apologize to the reviewer for not being able to identify the existing error. We ask that you indicate the error so that we can make the exact correction, please.
- Contribution to the field is omitted
Answer: As suggested, the following sentence has been inserted on page 14 line 312. “This article contributes to expanding the field of sports science, providing information that the analysis of the performance of professional soccer teams goes beyond psychological, technical, and tactical factors, by indicating that extra-field factors, such as the pandemic, directly impact sports performance.”
Reviewer 2 Report
Dear Authors,
the Authors did a good job in writing a readable manuscript. This paper could be considered qualified to be published on IJERPH if the authors apply the modifications requested, "Minor issues" are recommended. It is the reason why "Minor revision" is my personal choice.
The Authors aimed to verify if there was a difference in the home advantage in the points obtained between the last four seasons (2018, 2019, 2020, and 2021) of the Brazilian Serie A Championships. They also aimed to verify if there was a difference in the home advantage in the points obtained, the home advantage of the performance, and disciplinary aspects between the rounds without and with the crowd in the last seasons (2020 and 2021). Then, Authors aimed to describe the number of athletes absent according to the mandatory health protocol and we tested the association between home advantage in the points obtained and the number of athletes absent due to the health protocol.
While it is a very interesting topic, some suggestions for improving the paper are provided below:
· The basic descriptive level adopted by author(s) does not seem compatible with interests of both researchers' conceptual challenges and coaches' practical applications to training. The main concern refers to the weak/absent theoretical background; there is no perceivable research question or rationale for the study presented to readers. Author(s) do not consider any specific difficulties or other aspects of performing and playing soccer to minimally justify the comparison; on the other hand, the use of the descriptive knowledge obtained from the observed differences was not discussed in terms of applications for athletes and coaches. In short, the manuscript is limited to a description of the situation. A “practical suggestion” paragraph has to be included in the Discussion
· The Discussion is too extended, and it seems to be a revision of the literature. Most of the information can be moved from the Discussion and included in the Introduction
Finally, this manuscript could be considered qualified to be published if the authors apply the previous modifications, therefore “Minor revision” is recommended.
Author Response
Dear Reviewer, I thank you for your patience and appreciable comments that have been so helpful in improving the manuscript’s quality. Just below are responses to these comments. I have updated the version of my manuscript, so you can check the new version in which I have taken into account your comments. I hope you will agree that this new version is better than the previous one.
1-The basic descriptive level adopted by author(s) does not seem compatible with interests of both researchers' conceptual challenges and coaches' practical applications to training. The main concern refers to the weak/absent theoretical background; there is no perceivable research question or rationale for the study presented to readers.
Answer: We agree with the reviewer about the lack of background in the previous file. With the suggested changes in the discussion and in the introduction, we changed the structure and understood that the new version meets what was requested. We now present the concept of the home advantage and the factors that can explain this advantage. Among several factors, is the presence of the crowd. Ahead, we present that due to the Covid-19 pandemic, several leagues have developed health protocols that prohibited the presence of the crowd. While different studies showed the effect of participating in a championship partially without the presence of the crowd, there were no studies that indicated the effect of the Covid-19 pandemic on Brazilian soccer. Two years after the end of the strict health protocol, there are still no studies that indicate how the Brazilian championship was affected in the last two seasons, in general, each season or with the rounds without the presence of the crowd. With this writing logic, we understand that we have provided a reference for the reader to identify the need to present a study that would help in the understanding of what happened in the Brazilian championship.
2-Author(s) do not consider any specific difficulties or other aspects of performing and playing soccer to minimally justify the comparison; on the other hand, the use of the descriptive knowledge obtained from the observed differences was not discussed in terms of applications for athletes and coaches. In short, the manuscript is limited to a description of the situation. A “practical suggestion” paragraph has to be included in the Discussion
Answer: With the orientation to insert the practical suggestion, we provided it on page 14 line 314. “The present study can help coaches and staff by indicating that they should pay attention to the effect of external factors on the presence of the crowd. Once the presence of the crowd can be limited, there can be a reduction in the home advantage with that, the preparation of visiting teams can be reviewed. For example, in situations where there is no crowd, the home team has fewer offensive actions, such as corners or shots, which may allow the away team's command to organize its team with a more offensive tactic.”
3-The Discussion is too extended, and it seems to be a revision of the literature. Most of the information can be moved from the Discussion and included in the Introduction
Answer. We appreciate the indication of the need to shorten the intro, we ask that you review both the intro and the discussion and note the changes that have been made as indicated. With these changes, we believe that the introduction presents a more theoretical framework, and the discussion is better presented.